# The Effect of Ethephon on Ethylene and Chlorophyll in *Zoysia japonica* Leaves

**DOI:** 10.3390/ijms25031663

**Published:** 2024-01-29

**Authors:** Jiahang Zhang, Lijing Li, Zhiwei Zhang, Liebao Han, Lixin Xu

**Affiliations:** College of Grassland Science, Beijing Forestry University, Beijing 100083, China; 15104490515@163.com (J.Z.); llj15122428388@163.com (L.L.); zhangzw559@163.com (Z.Z.)

**Keywords:** *Zoysia japonica*, ethephon, ethylene, chlorophyll, non-stressed conditions, cold stress

## Abstract

*Zoysia japonica* (*Zoysia japonica* Steud.) is a kind of warm-season turfgrass with many excellent characteristics. However, the shorter green period and longer dormancy caused by cold stress in late autumn and winter are the most limiting factors affecting its application. A previous transcriptome analysis revealed that ethephon regulated genes in chlorophyll metabolism in *Zoysia japonica* under cold stress. Further experimental data are necessary to understand the effect and underlying mechanism of ethephon in regulating the cold tolerance of *Zoysia japonica*. The aim of this study was to evaluate the effects of ethephon by measuring the enzyme activity, intermediates content, and gene expression related to ethylene biosynthesis, signaling, and chlorophyll metabolism. In addition, the ethylene production rate, chlorophyll content, and chlorophyll *a*/*b* ratio were analyzed. The results showed that ethephon application in a proper concentration inhibited endogenous ethylene biosynthesis, but eventually promoted the ethylene production rate due to its ethylene-releasing nature. Ethephon could promote chlorophyll content and improve plant growth in *Zoysia japonica* under cold-stressed conditions. In conclusion, ethephon plays a positive role in releasing ethylene and maintaining the chlorophyll content in *Zoysia japonica* both under non-stressed and cold-stressed conditions.

## 1. Introduction

Cold stress affects the metabolism, growth, and development of plants, resulting in slow growth, yellowing, and necrosis of plants [1]. Photosynthesis is one of most susceptible biological processes to cold stress in C4 crops [2]. Therefore, identifying the regulatory mechanism to improve photosynthesis under cold stress will provide further insights into improving the cold resistance of plants.

Chlorophyll is mainly composed of chlorophyll *a* (Chl *a*) and chlorophyll *b* (Chl *b*) and plays a pivotal role in photosynthesis [3,4]. Chlorophyll can harvest and transduce light energy, and charge separation and electron transport in reaction centers; its content is also an important index for evaluating photosynthetic capacity [5,6]. The general process of chlorophyll biosynthesis includes three main steps: firstly, 5- aminolevulinic acid (ALA) is generated from glutamate, secondly, protoporphyrin IX is generated from ALA, and finally, chlorophyll is generated from protoporphyrin IX [7]. In the chlorophyll cycle, Chl *b* is synthesized from chlorophyllide *a* by chlorophyllide *a* oxygenase (CAO), Chl *b* is converted into 7-hydroxymethyl-chlorophyll *a* by Chl *b* reductase, Chl *b* reductase is encoded by NON-YELLOWCOLORING 1 (NYC1), and NYC1-LIKE (NOL) converts Chl *b* into 7-hydroxymethyl Chl *a*. Then, 7-hydroxymethyl Chl *a* is converted into Chl *a* under the catalysis of 7-hydroxymethyl Chl *a* reductase (HCAR) [8,9]. The main degradation mode of chlorophyll is through the ‘PAO (pheophorbide *a* oxygenase) pathway’ [10]. The first step in chlorophyll degradation is the removal of magnesium (Mg) from Chl *a* by a Mg-dechelatase called STAY GREENs (SGRs), which converts substrate into pheophytin *a* (Phein *a*) [11]. Then, Phein a is hydrolyzed by pheophytinase (PPH) to pheophorbide *a* (Pheide *a*) and phytol [12]. Subsequently, the porphyrin ring of Pheide *a* is cleaved by PAO and converted into red Chl catabolite (RCC) [13]. Finally, RCC is catalyzed by the red Chl actabolite reductase (RCCR) to form a primary fluorescent chlorophyll decomposition metabolite (pFCC), which is then transferred from chloroplasts and isomerized into non-fluorescent products through acidic pH in vacuole [14,15,16]. It has been found that cold stress affects both the biosynthesis and degradation of chlorophyll of plants and leads to a decrease in chlorophyll content, which may further affect the ability of chloroplasts to capture light signals [3].

Ethylene, which is a gaseous phytohormone, plays a key role in the regulation of photosynthetic characteristics (chlorophyll content, stomatal conductance, light dissipation, carbon fixation, and carbohydrate partitioning) [17]. The process of ethylene synthesis starts with conversion of S-AdoMet (SAM) into 1-aminocyclopropane-1-carboxylate (ACC) regulated by 1-aminocyclopropane-1-carboxylate synthase (ACS), followed by the conversion of ACC into ethylene by 1-aminocyclopropane-1-carboxylate oxidase (ACO), in a process called the Met cycle [18]. The ethylene signaling pathway starts from the ethylene receptor, which is a negative regulator of ethylene response, meaning that, in the presence of ethylene, the receptor is inactivated, which leads to ethylene signaling [19,20]. The ethylene receptor gene family in rice (*Oryza sativa* L. var. *Lansheng*) includes *Protein Kinase1* (*PK1*)/*ETR2*/*ETHYLENE RESPONSE2 like1* (*ERL1*), *PK2/ETR3*, *PK3/ETR4*, *ERS1*, and *ERS2* [21,22,23]. When the receptors perceive ethylene, the constitutive triple response (CTR1) is inactivated, which results in the cleavage and translocation of the ethylene insensitive 2 (EIN2) C-terminal part to the nucleus [24,25,26,27]. In the nucleus, the C-terminal part of EIN2 stabilizes ethylene insensitive 3 (EIN3) and EIN3-like proteins (EILs), preventing them from proteasomal degradation mediated by EIN3-binding F-box 1 (EBF1) and EIN3-binding F-box 2 (EBF2) [28,29]. EIN3 and EILs promote the expression of *ethylene response factor* (*ERF*) family genes, which are downstream regulators of ethylene response [30,31,32,33,34]. It has been reported that a low ethylene concentration facilitates the activation of defense signaling, while a high concentration inhibits the defense signaling of plants [35,36]. The application of ethephon could enhance the cold tolerance in plants by activating *ERF* genes [37,38]. Meanwhile, studies have shown the that overexpression of *TERF2*, an *ERF* family gene, reduces chlorophyll loss in rice (*Oryza sativa* L.) under cold stress [38]. However, the connection between ethylene and chlorophyll metabolism still needs to be studied.

*Zoysia japonica* (*Zoysia japonica* Steud.) is a kind of warm-season turfgrass with many excellent characteristics, including strong resistance to abiotic stress, disease, and insects, partly due to their well-developed rhizomes and stolons, making it widely used in warm climatic and transitional regions at ornamental venues, parks, sports field, soil and water conservation sites, and so on [39,40]. Because *Z. japonica* is adapted to a warm and humid climate, the shorter green period and longer dormancy caused by cold stress are the most limiting factors affecting its potential wider application [41,42,43]. As a direct ethylene source, ethephon is applied to plants and elicits a response identical to that induced by ethylene gas, for evaluating the impact of ethylene on plants [44,45]. Studies have confirmed that ethephon can indeed function as ethylene and improve the cold resistance of plants [46,47]. Moreover, in our previous studies, ethephon could reduce the chlorophyll loss of *Z. japonica* under cold stress, and a transcriptome analysis showed that ethephon regulated chlorophyll biosynthesis and degradation [47]. However, there is little research on the regulatory mechanism of how ethephon reduces the chlorophyll loss of *Z. japonica* under cold stress.

The aim of this research is to explore the role of ethephon in *Z. japonica* leaves under cold stress by analyzing the gene expression, enzyme activity, and intermediate substance content in ethylene biosynthesis, signaling, and chlorophyll metabolism, both at a normal temperature and under cold stress in *Z. japonica*. This study could provide a theoretical basis for extending the greening period and improving the turf quality of *Z. japonica* in late autumn through ethephon application.

## 2. Results

### 2.1. Effect of Ethephon Application on Ethylene Biosynthesis

#### 2.1.1. Effect of Ethephon Application on Ethylene Biosynthesis under Non-Stressed Conditions

Under non-stressed conditions, after 51 days of growth, the activity of ACO significantly increased, while the ethylene production rate significantly decreased (Figure 1). (Data for enzyme activity, intermediate content, and ethylene production rate in chlorophyll biosynthesis in different *Z. japonica* plants on 0 d and 52 d are in Appendix A). *ZjACO1* was up-regulated and *ZjSAMS3*, *ZjACS*, *ZjACO*, and *ZjACO1X1* were down-regulated (Figure 2) (Data for the relative expression of ethylene synthesis genes in different *Z. japonica* plants are in Appendix A).

Under non-stressed conditions, ethephon application inhibited the activity of ACS and ACO, which are two key enzymes involved in ethylene biosynthesis, and decreased the content of ACC, the ethylene precursor (Figure 3). However, ethephon application inhibited the decrease in the magnitude of the ethylene production rate in *Z. japonica* under non-stressed conditions (Figure 1). Overall, ethephon application inhibited ethylene biosynthesis enzyme (ACO and ACS) activity but stimulated some of the ethylene-synthesis-related genes (up-regulating *ZjACO* and *ZjACO1X1*, and down-regulating *ZjACO1*) (Figure 2). Ethephon increased the ethylene production rate in *Z. japonica* under non-stressed conditions relative to the non-ethephon-treated control group (NT 52 d), showing that ethephon can release ethylene without relying on endogenous ethylene biosynthesis (Figure 1 and Figure 3).

#### 2.1.2. Effect of Ethephon Application on Ethylene Biosynthesis under Cold Stress

Cold stress decreased the activity of ACO and, eventually, the ethylene production rate significantly (Figure 1). Cold stress impacted the expression of ethylene-synthesis-related genes (by up-regulating *ZjACS* and *ZjACO*, and down-regulating *ZjACO1* and *ZjACO1X1*) (Figure 2) in *Z. japonica*. Ethephon application decreased the activity of ethylene biosynthesis enzymes (ACO) (Figure 3) and stimulated ethylene biosynthesis genes (by up-regulating *ZjACO1* and *ZjACO1X1*, and down-regulating *ZjSAMS3*, *ZjACS*, and *ZjACO*) (Figure 2). Ethephon repressed the decreasing magnitude of the ethylene production rate (Figure 1 and Figure 3), showing that ethephon can also release ethylene without relying on ethylene biosynthesis in *Z. japonica* under cold stress.

### 2.2. Effect of Ethephon Application on Ethylene Signaling under Non-Stressed Conditions and Cold Stress

The decrease in the ethylene production rate in *Z. japonica* under non-stressed conditions (Figure 1) stimulated ethylene signaling genes by down-regulating all the detected ethylene signaling genes in this research (Figure 4) (Data for the relative expression of ethylene signaling genes in different *Z. japonica* plants are in Appendix A). In contrast, the increase in the ethylene production rate in *Z. japonica* caused by ethephon application under non-stressed conditions (Figure 1) up-regulated *ZjETR3X1*, *ZjCTR1X1*, *ZjEIL1-2*, and *ZjEIL3* and down-regulated *ZjEBF1X1* relative to that in *Z. japonica* under non-stressed conditions only (Figure 4).

The decrease in the ethylene production rate in *Z. japonica* under cold stress (Figure 1) stimulated ethylene signaling genes by up-regulating *ZjETR3X1*, *ZjEBF1X1*, *ZjEIL1-1*, *ZjEIL1-2*, and *ZjEIL3*, while down-regulating *ZjERS2X1*, *ZjCTR1X1*, and *ZjEIN2X1* (Figure 4). Meanwhile, the increase in the ethylene production rate in *Z. japonica* caused by ethephon application under cold stress (Figure 1) up-regulated *ZjCTR1X1*, *ZjEIN2X1*, and *ZjEIL1-1* and down regulated *ZjETR3X1*, *ZjEBF1X1*, *ZjEIL1-2*, and *ZjEIL3* compared with that in *Z. japonica* under cold stress only (Figure 4).

### 2.3. Effect of Ethephon Application on Chlorophyll Metabolism

#### 2.3.1. Effect of Ethephon Application on Chlorophyll Metabolism under Non-Stressed Conditions

Porphobilinogen (PBG) content significantly decreased and PBGD activity significantly increased under non-stressed conditions after 51 days of natural growth (Figure 5) (Data for enzyme activity and intermediate content in chlorophyll metabolism are in Appendix A). PBG was converted into chlorophyll precursors with increasing activity of PBGD (Figure 6). Chlorophyll synthesis was promoted by increasing the activity of chlorophyll synthetases (PBGD) and stimulated chlorophyll synthesis genes (by up-regulating *ZjPBGD*, *ZjUROS*, *ZjCHLH*, *ZjCHLI*, *ZjPORA*, *ZjPORB*, and *ZjCAO*, and down-regulating *ZjURO1*, *ZjURO2*, *ZjHEMF*, *ZjCHLD*, *ZjCHLI*, and *ZjCHLM*) (Figure 7) (Data for the relative expression of chlorophyll metabolism genes in different *Z. japonica* plants are in Appendix A). Meanwhile, the activity of PPH and RCCR was significantly increased (Figure 5), and the genes in the process of chlorophyll breakdown detected in this research were stimulated (by down-regulating *ZjNYC1*, *ZjHCAR*, *ZjCLH2*, *ZjPAO*, and *ZjRCCR*) (Figure 7) under non-stressed conditions after 51 days. With both promotion in synthesis and degradation, the chlorophyll content and Chl *a*/*b* did not undergo significant changes under non-stressed conditions after 51 days (Figure 8) (Data for chlorophyll content and Chl *a*/*b* ratio in different *Z. japonica* plants on 0 d and 52 d are in Appendix A).

Ethephon application in *Z. japonica* under non-stressed conditions led to an increase in ALAD activity and the inhibition of a decrease in PBG content in *Z. japonica* compared to *Z. japonica* under non-stressed conditions only (Figure 5). However, the decrease in PBGD activity meant that the conversion of PBG into chlorophyll precursor substances was inhibited, which meant that the chlorophyll synthesis was inhibited by regulating the activity of chlorophyll synthesis enzymes (ALAD and PBGD) (Figure 6) and stimulating chlorophyll synthesis genes (by up-regulating *ZjURO1*, *ZjHEMF*, *ZjCHLH*, *ZjPORA*, and *ZjCAO*, and down-regulating *ZjPBGD*, *ZjUROS*, *ZjURO2*, *ZjCHL1*, and *ZjPORB*) (Figure 7) in *Z. japonica* with ethephon application under non-stressed conditions compared to *Z. japonica* under non-stressed conditions only. Interestingly, ethephon application inhibited the chlorophyll breakdown in *Z. japonica* under non-stressed conditions by significantly decreasing the activity of all enzymes investigated related to chlorophyll breakdown (Figure 5 and Figure 6), and ethephon up-regulated *ZjPAO* and *ZjRCCR* (Figure 7). However, ethephon application did not significantly change the chlorophyll content and Chl *a*/*b* in *Z. japonica* under non-stressed conditions (Figure 8).

#### 2.3.2. Effect of Ethephon Application on Chlorophyll Metabolism under Cold Stress

Cold stress did not significantly change the activity of enzymes and the content of intermediate substances involved in chlorophyll synthesis (Figure 5), while it changed the expression level of chlorophyll synthesis genes (by up-regulating *ZjPBGD*, *ZjURO1*, *ZjCHLD*, and *ZjCAO*, and down-regulating *ZjUROS*, *ZjURO2*, *ZjHEMF*, *ZjCHLH*, *ZjCHLI*, *ZjCHLM*, *ZjPORA*, and *ZjPORB*) (Figure 7) in *Z. japonica* after 51 days. Cold stress significantly increased the activity of MDCase, a key enzyme involved in chlorophyll breakdown (Figure 6), and regulated the expression of chlorophyll-degradation-related genes (by up-regulating *ZjNYC1*, *ZjCLH2*, *ZjPAO*, and *ZjRCCR*, and down-regulating *ZjHCAR*) (Figure 7) in *Z. japonica* after 51 days, showing that the chlorophyll breakdown was promoted on a gene expression level. Meanwhile, cold stress significantly decreased the chlorophyll content and Chl *a*/*b* in *Z. japonica* after 51 days (Figure 8).

Ethephon application significantly decreased PBGD activity (Figure 5) and changed the expression of genes in chlorophyll synthesis (by up-regulating *ZjHEMF* and *ZjChlM*, and down-regulating *ZjPBGD*, *ZjURO1*, *ZjCHLD*, *ZjCHLH*, *ZjPORA*, and *ZjCAO*) (Figure 7), showing that chlorophyll synthesis was mainly inhibited. In the process of chlorophyll breakdown, ethephon application decreased the activity of MDCase, increased the activity of PAO (Figure 6), and changed the expression of chlorophyll breakdown genes (by down-regulating *ZjNYC1*, *ZjCLH2*, and *ZjPAO*) (Figure 7) in *Z. japonica*, showing that chlorophyll breakdown was mainly inhibited. Meanwhile, ethephon application inhibited the decrease in chlorophyll content and increased Chl *a*/*b* in *Z. japonica* under cold stress (Figure 8), which indicated that ethephon application promoted chlorophyll content and Chl *a*/*b* in *Z. japonica* under cold stress mainly by inhibiting chlorophyll breakdown.

## 3. Discussion

### 3.1. Ethephon Plays a Positive Role in Ethylene Releasing and Ethylene Signaling and It May Regulate Ethylene Biosynthesis in a Negative Feedback Loop in Z. japonica Leaves

The regulation of ethylene biosynthesis between different organs in higher plants mainly depends on the operation of ACC [48]. Shi et al. [49] found that the ethylene production rate in leaves of *Paeonia suffruticosa* decreased gradually from flowering to senescence, and the percentage of ethylene production in the total organ decreased gradually. The ethylene production rate in petals gradually increased, and the percentage of ethylene production in the total organ gradually increased, indicating that organs other than petals regulated the ethylene production by transferring ACC to petals, thus affecting the flowering and senescence process. Research has found that it is especially important to maintain low ethylene levels during vegetative development in order to optimize the growth process of roots and shoots [50]. In this research, the ACS activity and ACC content increased slightly, though without significant changes. Therefore, we speculate that the leaves of *Z. japonica* promote ACC content by increasing ACS activity and transporting it to the roots to promote their growth, ultimately leading to a decrease in the ethylene production rate in the leaves. Ethylene has also been shown to directly regulate its own biosynthesis by controlling ACS and ACO expression in negative ERF-mediated feedback mechanisms [50,51,52]. In agreement, there might be the same negative feedback mechanism of ethylene synthesis in *Z. japonica*.

Previous studies have shown that ethephon can spontaneously release ethylene without relying on ACC as a substrate [53,54,55], which is consistent with our findings. An increase in the ethylene production rate in *Z. japonica* under non-stressed conditions caused by ethephon up-regulates upstream genes of *ERFs* (*ZjEIL1-2* and *ZjEIL3*), down regulates *ZjACO1*, and decreases ACO activity, ultimately affecting ethylene biosynthesis in *Z. japonica* under non-stressed conditions.

It was found that cold stress could promote ethylene production in various plants [56,57,58], and ethylene could induce the antioxidative defense system in plants, which reduces oxidative stress and maintains growth and photosynthetic efficiency [59]. However, previous research has suggested that ethephon treatment stimulated ethylene production just at the initial stage of cold storage, and subsequently decreased over time [60,61]. Shutting down stress-induced ethylene biosynthesis is also important, as excessive defense reactions can hinder plant growth and survival [52]. Therefore, we speculate that cold stress can promote ethylene production in the initial stage, but, after a long time, likewise in our study, 51 days, an increase in ethylene production can activate the negative feedback loop of ethylene biosynthesis itself, that is, an increase in ethylene production increases the expression of *ZjEIL1-1*, *ZjEIL1-2*, and *ZjEIL3* involved in ethylene signaling, thus inhibiting the expression of *ZjACO1* and *ZjACO1X1*, resulting in decreased ACO activity to further inhibit ethylene biosynthesis in *Z. japonica* under cold stress.

Ethephon induces a rapid increase in ethylene in plant leaves [46]; in our study, the ethylene production rate remained higher in *Z. japonica* with ethephon application after 51 d of cold stress. It has been found that the EIN2 protein is the central positive regulator of the ethylene signaling pathway, and EIN3 and EIL1 are the key transcription factors downstream of EIN2 [62,63]. Meanwhile, studies have indicated that the ethylene signaling pathway down-regulates ethylene biosynthesis in a negative feedback manner via repressing *ACS2/ACS6* expression through *EIN2*, *EIN3*, and *EIL1* indirectly in Arabidopsis [52]. Therefore, the rapid increase in ethylene production in *Z. japonica* due to ethephon application under cold stress may activate the negative feedback loop of ethylene by up-regulating *ZjEIN2X1* and *ZjEIL1-1*, thus inhibiting ethylene biosynthesis by down-regulating *ZjACS* and *ZjACO* and decreasing ACO activity in *Z. japonica* under cold stress. Taken together, the above results indicate that ethephon induces ethylene release and ethylene itself, as well as ethylene signaling, which may regulate ethylene biosynthesis in a negative feedback loop in *Z. japonica* under cold stress.

### 3.2. Ethephon-Induced Ethylene Plays a Positive Role in Maintaining Chlorophyll Content in Z. japonica Leaves

It has been found that ethylene-mediated growth inhibition results in smaller leaves because of restricted cell expansion, and it favors overall plant growth and stress tolerance [34]. Consistently, we observed the same effect in our research, in that ethephon application inhibited leaf width growth but resulted in the overall growth of the plants (Figure 9). The chlorophyll content depends on the relative rates of chlorophyll synthesis and breakdown [64,65]. Our results showed that ethephon application in *Z. japonica* under non-stressed conditions contributed to maintaining the chlorophyll content. We propose that ethephon application improves chlorophyll content by inhibiting chlorophyll synthesis (by inhibiting PBGD activity) and breakdown (by inhibiting the activity of CHLase, MDCase, PPH, PAO, and RCCR) in *Z. japonica* under non-stressed conditions. Previous studies have found that ethylene-mediated signaling was involved in the maintenance of chlorophyll content and enhancement of thermotolerance in plants under stress [66]. In this research, ethephon application under cold stress increased the chlorophyll content by inhibiting chlorophyll synthesis (by inhibiting PBGD activity) and vastly inhibiting chlorophyll breakdown (by inhibiting MDCase activity). Therefore, we speculate that ethephon application improves the plant growth of *Z. japonica* both under non-stressed conditions and under cold stress in an energy saving way, reducing both synthesis and degradation.

PBG deaminase (PBGD), also known as hydroxymethylbilane synthase, involves the deamination and polymerization of four molecules of the monopyrrole PBG, resulting in a highly unstable 1-hydroxymethylbilane (preuroporphyrinogen), which is necessary for the formation of chlorophyll and heme in plant cells [67,68]. Possible benefits for this PBG accumulation may be that it can converted into a variety of intermediary metabolites in addition to porphyrins [69]. This metabolism involves reactions which are partially dependent upon O_2_ and pyridoxal phosphate. Moreover, accumulated porphyrin precursors, including PBG, might induce defense responses such as the reactive oxygen species (ROS) scavenging mechanism [70,71]. Therefore, it is speculated that ethephon application induces the ROS scavenging mechanism by inhibiting PBGD activity both under non-stressed conditions and cold stress in *Z. japonica*.

The Chl *a* and Chl *b* molecules are integral components of the light-harvesting complex. Chl *a* can be degraded into colorless fluorescent substances by a series of chlorophyll-degrading enzymes, mainly chlorophyllase (CLH), pheophytinase (PPH), and Mg-dechelatase (MDCase) [72]. Chl *b* is a major photosynthetic pigment in green plants that is synthesized by chlorophyllide a oxygenase (CAO), which is partly regulated on a transcriptional level by the expression of the CAO gene. Our results showed that ethephon application in *Z. japonica* under cold stress increased the Chl *a*/*b* ratio. This process may be achieved by inhibiting the degradation of Chl *a* (decreasing MDCase activity) and the synthesis of Chl *b* (down-regulating *ZjCAO*). In the presence of H_2_O_2_, Chl *a* can be converted into 132-hydroxychlorophyll *a* to promote chlorophyll breakdown [73]. Meanwhile, ROS further aggravate chlorophyll breakdown by directly attacking the structure of the chlorophyll pyrrole ring and destroying the carbon ring double bond and porphyrin macroring [74]. Therefore, we speculate that the degradation of Chl *a* through inhibiting MDCase activity in *Z. japonica* caused by ethephon application might help in reducing ROS in *Z. japonica* under cold stress. The biosynthesis and regulation of Chl *b* play important roles in adjusting the antenna size of photosystems [75]. Also, the relative concentrations of Chl *a* and Chl *b* are used as indicators of the antenna size. Maximizing light capture through the adjustment of antenna size can optimize light capture and light energy conversion [76]. Ethephon application in *Z. japonica* under cold stress increased the Chl *a*/*b* ratio, indicating that the antenna size may have been decreased. Previous studies have reported that the same adjustment of light-harvesting pigment in plants with a slightly reduced antenna size improved the photosynthetic performance in plants (by improving light capture efficiency, decreasing energy loss, and mitigating photodamage) [77,78]. Therefore, we propose that the increased Chl *a*/*b* ratio in *Z. japonica* under cold stress, through inhibiting the degradation of Chl *a* (decreasing MDCase activity) and inhibiting the synthesis of Chl *b* (down-regulating *ZjCAO*) caused by ethephon application, is probably still able to meet the need of light capture, meanwhile reducing the light-harvesting antenna size, thus ameliorating the loss of light energy and suppressing phototoxicity in *Z. japonica* under cold stress.

Ethephon application in *Z. japonica* under cold stress promoted PAO activity. Deficiency in PAO results in an accelerated cell death phenotype, which is caused by the accumulation of substrates of respective reactions, Pheide *a*. These colored intermediates of chlorophyll breakdown are potentially phototoxic, and tight control of the PAO pathway has been considered to be important for preventing premature cell death [79]. Therefore, we speculate that ethephon application may suppress phototoxicity through increasing PAO activity in *Z. japonica* under cold stress.

EIN3/EIL1 can prevent the photooxidation of etiolated cotyledons by regulating a number of tetrapyrrole pathway genes to inhibit the accumulation of protochlorophyllide, a phototoxic intermediate in chlorophyll synthesis [80,81]. Our results showed that ethephon application in *Z. japonica* under cold stress up-regulated *ZjEIL1-1* compared to *Z. japonica* under cold stress only. Therefore, we propose that ethephon application may inhibit phototoxicity by decreasing the excessive accumulation of protochlorophyllide in chlorophyll synthesis in *Z. japonica* under cold stress. However, further research is needed to investigate how *ZjEIL1-1* works in this process.

Our study indicated that the chlorophyll content of *Z. japonica* could be regulated not only by the genes in the chlorophyll synthesis and degradation pathway, but also by genes in the ethylene synthesis and signaling pathway, both at normal temperatures and under cold stress.

## 4. Materials and Methods

### 4.1. Plant Materials and Growth Conditions

*Zoysia japonica* Steud. cv. *Chinese Common* plants were obtained from the White Horse Experimental Ground of Nanjing Agricultural University (LAT 32°2′6.25″ N; LON 118°50′23.47″ E) on January 2019. The experiment was designed as a multiple-variable experiment. The variables included temperature and ethephon application. A total of twenty pots of plants were divided into four treatment groups (five pots per group): NT, normal day/night temperature (32/28 °C) with water pretreatment as a control; NE, normal day/night temperature (32/28 °C) with ethephon pretreatment (150 mg/L); CS, day/night cold stress (6/4 °C) with water pretreatment; and CE, day/night cold stress (6/4 °C) with ethephon pretreatment (150 mg/L). Except for different temperatures, the plants of the four treatment groups had the same environmental factors and were placed in the artificial climate room (RXZ-500C) for maintenance and management. The light period was 14 h and the light intensity was 17,600 lux. The ethephon pretreatment was carried out on 26 September 2019 (0 d), while the cold stress treatment was started on 27 September 2019 (1 d) (Figure 10). On 17 November 2019 (52 d), significant phenotypic differences were observed in the leaves of *Z. japonica* in each treatment group, so the treatment was stopped. Fresh leaves of each group were collected for the next experiment and stored at −80 °C after freezing in liquid nitrogen.

### 4.2. Determination of Chlorophyll Content in Leaves

The chlorophyll content was determined through ethanol extraction and it was slightly modified based on Li’s study [82]. Fresh leaves (approximately 0.05 g per sample) were cut into small sections of approximately 5 mm and placed into centrifuge tubes filled with 8 mL of 95% ethanol. The tubes were stored in the dark for 48 h, and the absorbance was measured at 665 nm and 649 nm [83]. The chlorophyll content was calculated according to the following formula:(1)$$C_{a}\;=13.95A_{665}\;-\;6.88A_{649},\;$$
(2)$$C_{b}\;=24.96A_{649}-7.32A_{665},$$
(3)$$C_{\mathrm{chl}}\;=C_{a}\;+C_{b},$$
(4)$$Chl\;a\;content=C_{a}\;\times \frac{V}{W},$$
(5)$$Chl\;b\;content=C_{b}\;\times \frac{V}{W},$$
(6)$$Chl\;content=C_{\mathrm{chl}}\;\times \frac{V}{W}$$

In the formula, C*_a_* refers to the concentration of Chl *a* (mg/L), C*_b_* refers to the concentration of Chl *b* (mg/L), C_Chl_ refers to the total chlorophyll concentration (mg/L), V refers to the volume after extraction (L), W refers to the sample weight (g), and A_665_ and A_649_ refer to the absorbance at 665 nm and 649 nm.

### 4.3. Method for Determination of Ethylene Production Rate

The ethylene production rate in the plants was determined according to Liu’s method [84]. After the leaves were immersed in distilled water for 24 h (to avoid ethylene production from the wound), they were transferred to a 10 mL sealed bottle containing 1 mL of distilled water and placed in the dark at room temperature for 24 h to collect ethylene. Then, the ethylene absorption peak area of the top gas extracted using a sampling needle was measured using a gas chromatograph (GC17A, Shimadzu, Kyoto, Japan) [85]. The standard curves of the absorption peak area of ethylene production at different gradients were drawn to calculate the ethylene production of each treatment group, and finally, the ethylene production rate of each treatment group was obtained. We used a change rate of the ethylene production rate to calculate the ethylene production rate before treatment (0 d) and after treatment (52 d), which indicated an increase (>1) or a decrease (<1) in the ethylene production rate after treatment.
(7)$$\mathrm{the}\;\mathrm{change}\;\mathrm{rate}\;\mathrm{of}\;\mathrm{ethylene}\;\mathrm{production}\;\mathrm{rate}\;=\frac{ethylene\;production\;rate\;on\;52\;d}{ethylene\;production\;rate\;on\;0\;d}$$

### 4.4. Determination of Enzyme Activities

The enzyme activities (ACS, ACO, MDcase, RCCR, PAO, PPH, CLH, and PBGD) were measured by Shanghai Jianglai Biotechnology Co., Ltd., Shanghai, China (http://shjlswkj3283636.yixie8.com/) (accessed on 27 December 2023) through double-antibody sandwich enzyme-linked immunosorbent assay (ELISA) [86]. Purified enzymes were used to capture antibodies and were coated on a microplate to produce solid-phase antibodies. The pre-purified substances under test were added to the corresponding wells of an ELISA plate, and they were then combined with HRP-labeled detection antibodies to obtain a complex. After thorough washing, the complex was colored with TMB substrate. Through the catalysis of HRP enzyme, TMB was transformed, appeared blue, and was ultimately converted into a yellow substance under acidity. The depth of color was positively correlated with the activity in the tested leaves. The absorbance of each sample at 450 nm was measured with a Rayto RT-6100 enzymoleter and the ACS, ACO, MDcase, RCCR, PAO, PPH, CLH, and PBGD activities were calculated according to the standard curves.

The determination of ALAD activity was referred to Mauzerall’s method [87]. In total, 0.1 g of leaf tissue was cut into pieces and put into a mortar. A total of 1.8 mL of extraction buffer (0.1 M potassium phosphate (pH 7.8), 8 mM magnesium chloride, 0.5% Triton X-100, 30 g/L polyvinylpyrrolidone, and 1 mM PMSF) was added into a mortar and ground on ice. The ground materials were transferred into a centrifuge tube, centrifuged with a centrifuge (FRESCO 21) of 9000× *g* for 15 min, and the supernatant was obtained. In total, 200 µL of enzyme extract was sucked into a new centrifuge tube, 300 µL of 50 mM ALA was added, and after 30 min of reaction at 37 °C, 1 mL of 100 g/L TCA was added to stop the reaction. An equal volume (600–800 µL) of Ehrlich reagent was added into the supernatant (600–800 µL) and treated in dark for 15 min, and the absorbance was measured at 553 nm with a spectrophotometer to calculate the content of PBG. The enzyme activity unit of an ALAD was 1μmol of PBG per hour catalyzed by no milligram of protein under the above reaction conditions. We used the enzyme activity change rate to calculate the enzyme activity before treatment (0 d) and after treatment (52 d), which indicated an increase (>1) or a decrease (<1) in the enzyme activity after treatment.
(8)$$\mathrm{enzyme}\;\mathrm{activity}\;\mathrm{change}\;\mathrm{rate}\;=\frac{\mathrm{enzyme}\;\mathrm{activity}on\;52\;d}{\mathrm{enzyme}\;\mathrm{activity}on\;0\;d}$$

### 4.5. Determination of the Ethylene Precursor and Chlorophyll Precursor

The ACC content was measured by Shanghai Jianglai Biotechnology Co., Ltd., Shanghai, China (http://shjlswkj3283636.yixie8.com/) (accessed on 27 December 2023) through double-antibody sandwich enzyme-linked immunosorbent assay (ELISA) [86]. The absorbance of each sample at 450 nm was measured with a Rayto RT-6100 enzymoleter and the ACC content was calculated according to the standard curve.

The determination of PBG content refers to Bogorad’s method [88]. In total, 0.1 g of fresh leaves were cut into pieces and 2 mL of extraction buffer was added (0.6 mol/L Tris-HCL, 0.1 mol/L EDTA, pH 8.2). The leaves in the mortars were ground on ice until they were homogenized, then they were transferred to the centrifugal tube, centrifuged with a centrifuge (FRESCO 21) of 12,000× *g* for 15 min, and the supernatant was sucked into a new tube. Then, an equal amount of Ehrlich reagent (2% p-dimethylaminobenzaldehyde, 6% perchloric acid, 88% acetic acid) was added to the supernatant and reacted in the dark for 15 min. The absorbance at the wavelength of 553 nm was measured using a spectrophotometer, and the PBG content (μmol/g) was calculated by:

MEC(molar extinction coefficient) = 6.1 × 10^4^ L/mol·cm(9)

We used the change rate of the ethylene precursor content and chlorophyll precursor content to calculate the precursor content before treatment (0 d) and after treatment (52 d), which indicated an increase (>1) or a decrease (<1) in the precursor content after treatment.
(10)$$\mathrm{precursor}\;\mathrm{content}\;\mathrm{change}\;\mathrm{rate}\;=\frac{\mathrm{precursor}\;\mathrm{content}on\;52\;d}{\mathrm{precursor}\;\mathrm{content}on\;0\;d}$$

### 4.6. Methods for Determination of Relative Gene Expression

The TaKaRa MiniBEST Plant RNA Extraction Kit (TaKaRa, Japan-b) (Beijing, China) was used to extract the total RNA of *Z. japonica* through grinding, splitting, cleaning, digestion, elution, and other steps, while the PrimeScriptTMRT reagent Kit with the gDNA Eraser (Takara, Japan-a) (Beijing, China) was used to synthesize the cDNA and remove the genomic DNA for a subsequent determination of gene expression. The primers were designed and synthesized by RuiBiotech Co., Ltd., Beijing, China (http://www.ruibiotech.com/) (accessed on 27 December 2023). The instructions for the TB Green Premix Ex Taq kit (Tli RNaseH Plus) were obtained from Takara Biomedical Technology (Beijing) Co., Ltd., Beijing, China (https://www.takarabiomed.com.cn/) (accessed on 27 December 2023), and qRT-PCR was performed using a Bio Rad CFX Connect Real-Time PCR Detection System (Bio-Rad Laboratories Co., Ltd., CA, USA) (https://www.bio-rad.com/) with the following parameter settings: 95 °C for 30 s, followed by 40 cycles of 95 °C for 5 s and annealing/extension at 60 °C for 30 s. Single-product amplification was confirmed with a melt curve. Relative gene expression was based on the 2^−ΔΔCT^ method [89], which is a convenient way to analyze the relative changes in gene expression from real-time quantitative PCR experiments.

### 4.7. Statistical Analysis

All experiments were repeated at least three times, and the data analysis was performed using IBM SPSS Statistics 20. The Levene test was used for comparison among different samples on day 0 and day 52. The difference was considered to be statistically significant when *p* < 0.05.

## 5. Conclusions

Ethephon application in a proper concentration can promote ethylene release, which may regulate ethylene biosynthesis in a negative feedback loop in *Z. japonica* through up-regulating *ZjEIL1-2* and *ZjEIL3*, down-regulating *ZjACO1*, and decreasing ACO activity under non-stressed conditions. Also, ethephon-induced ethylene also plays a positive role in maintaining chlorophyll content in *Z. japonica* by inhibiting chlorophyll synthesis (by inhibiting PBGD activity) and breakdown (by inhibiting the activity of CHLase, MDCase, PPH, PAO, and RCCR) under non-stressed conditions. Likewise, the negative feedback loop in *Z. japonica* under cold stress caused by ethephon application is achieved through up-regulating *ZjEIN2X1* and *ZjEIL1-1*, thus inhibiting ethylene biosynthesis by down-regulating *ZjACS* and *ZjACO* and decreasing ACO activity. Ethephon-induced ethylene also plays a positive role in maintaining the chlorophyll content in *Z. japonica* by inhibiting chlorophyll synthesis (by inhibiting PBGD activity) and chlorophyll breakdown (by inhibiting MDCase activity) under cold stress. Ethephon application in *Z. japonica* under non-stressed conditions and under cold stress improves plant growth, possibly through restricting cell expansion and suppressing phototoxicity in *Z. japonica*. Moreover, there may be some connection between ethylene signaling genes and chlorophyll metabolism genes, where *ZjEIL1-1* may be the key gene.

## Figures and Tables

**Figure 1 ijms-25-01663-f001:**
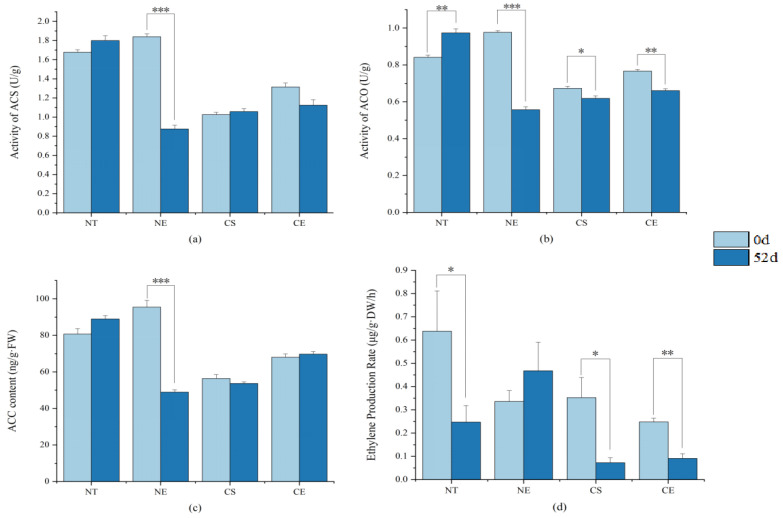
Enzyme activity, intermediate content, and ethylene production rate in ethylene biosynthesis in different *Z. japonica* plants on 0 d and 52 d; (**a**) ACS activity on 52 d compared to 0 d in different *Z. japonica* plants; (**b**) ACO activity on 52 d compared to 0 d in different *Z. japonica* plants; (**c**) ACC content on 52 d compared to 0 d in different *Z. japonica* plants; and (**d**) ethylene production rate on 52 d compared to 0 d in different *Z. japonica* plants; ’*’, ‘**’, and ‘***’ indicate significant differences at *p* < 0.05; *p* < 0.01; and *p* < 0.001, respectively. NT (control) refers to normal day/night temperatures (32/28 °C) with water pretreatment as a control; NE refers to normal day/night temperatures (32/28 °C) with ethephon pretreatment (150 mg/L); CS refers to day/night cold stress (6/4 °C) with water pretreatment; and CE refers to day/night cold stress (6/4 °C) with ethephon pretreatment (150 mg/L).

**Figure 2 ijms-25-01663-f002:**
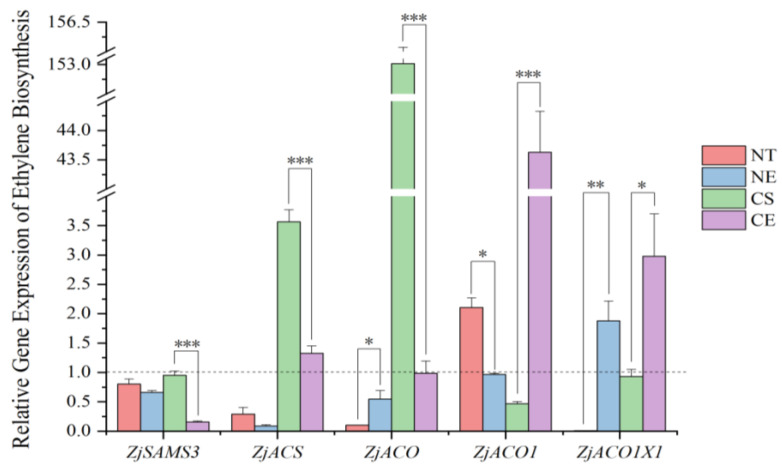
Relative gene expression in the process of ethylene biosynthesis in different *Z. japonica* plants on 52 d to 0 d. ‘*’, ‘**’, and ‘***’ indicate significant differences at *p* < 0.05; *p* < 0.01; and *p* < 0.001, respectively. The dashed line refers to the relative gene expression is 1.

**Figure 3 ijms-25-01663-f003:**
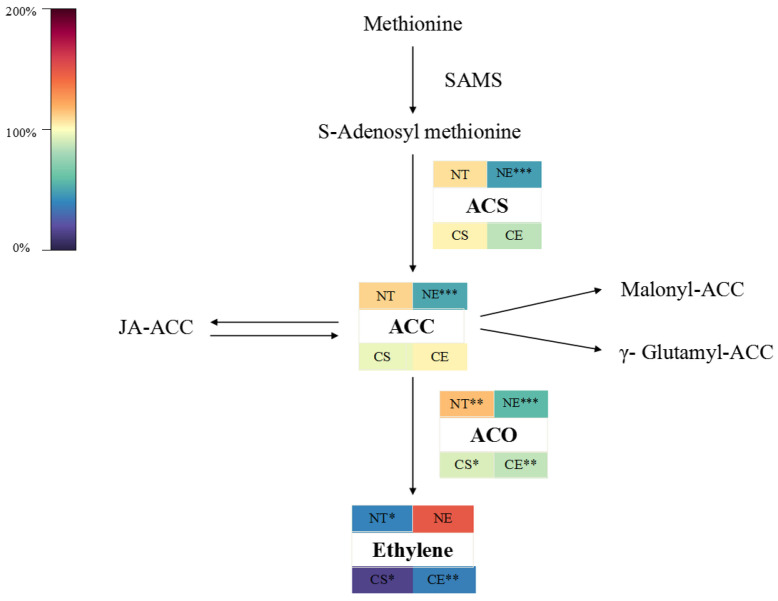
Heat map of change rate of enzyme activity, intermediate product content in ethylene biosynthesis, and ethylene production rate on 52 d compared to 0 d in different *Z. japonica* plants. ‘*’, ‘**’, and ‘***’ indicate significant differences at *p* < 0.05; *p* < 0.01; and *p* < 0.001, respectively.

**Figure 4 ijms-25-01663-f004:**
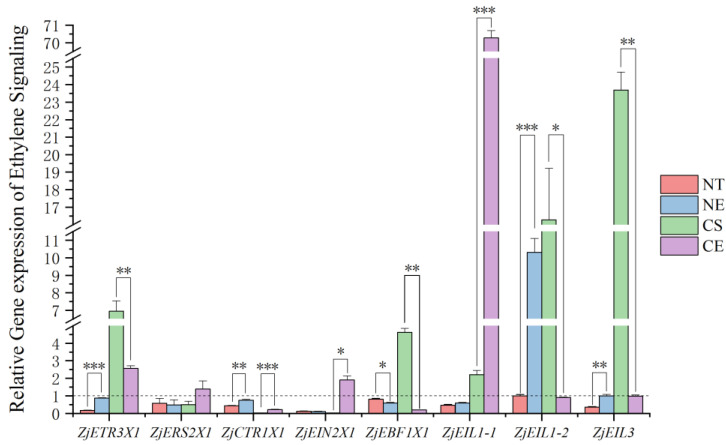
Relative gene expression in the process of ethylene signaling pathway in different *Z. japonica* plants on 52 d to 0 d. ‘*’, ‘**’, and ‘***’ indicate significant differences at *p* < 0.05; *p* < 0.01; and *p* < 0.001, respectively. The dashed line refers to the relative gene expression is 1.

**Figure 5 ijms-25-01663-f005:**
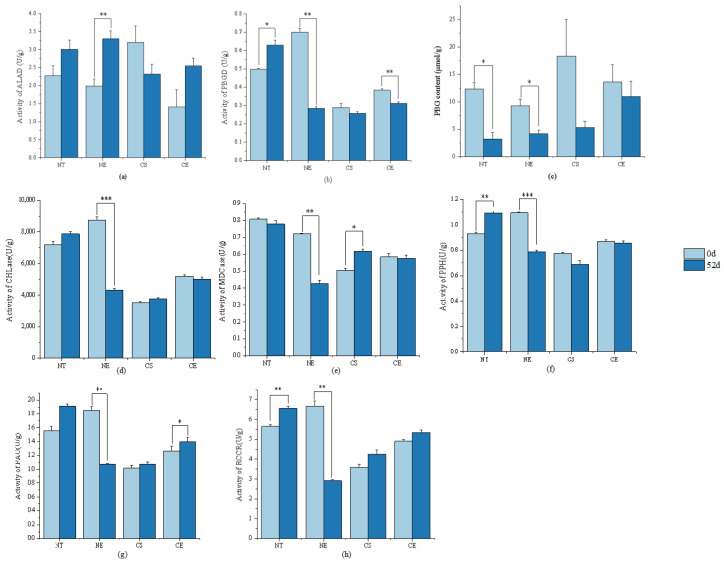
Enzyme activity and intermediate content in chlorophyll biosynthesis in different *Z. japonica* plants on 0 d and 52 d; (**a**) ALAD activity on 52 d compared to 0 d in different *Z. japonica* plants; (**b**) PBGD activity on 52 d compared to 0 d in different *Z. japonica* plants; (**c**) PBG content on 52 d compared to 0 d in different *Z. japonica* plants; (**d**) CHLase activity on 52 d compared to 0 d in different *Z. japonica* plants; (**e**) MDCase activity on 52 d compared to 0 d in different *Z. japonica* plants; (**f**) PPH activity on 52 d compared to 0 d in different *Z. japonica* plants; (**g**) PAO activity on 52 d compared to 0 d in different *Z. japonica* plants; and (**h**) RCCR activity on 52 d compared to 0 d in different *Z. japonica* plants; ‘*’, ‘**’, and ‘***’ indicate significant differences at *p* < 0.05; *p* < 0.01; and *p* < 0.001, respectively. NT (control) refers to normal day/night temperatures (32/28 °C) with water pretreatment as a control; NE refers to normal day/night temperatures (32/28 °C) with ethephon pretreatment (150 mg/L); CS refers to day/night cold stress (6/4 °C) with water pretreatment; and CE refers to day/night cold stress (6/4 °C) with ethephon pretreatment (150 mg/L).

**Figure 6 ijms-25-01663-f006:**
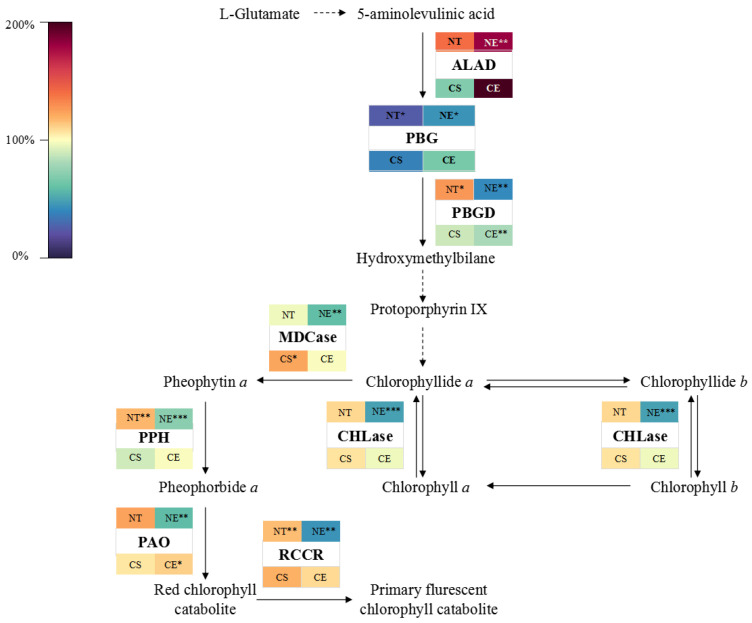
Heat map of change rate of chlorophyll metabolic enzyme activities and intermediate product content on 52 d compared to 0 d in different *Z. japonica* plants. ‘*’, ‘**’, and ‘***’ indicate significant differences at *p* < 0.05, *p* < 0.01, and *p* < 0.001, respectively, comparing values on 52 d to 0 d. ⇾ indicates synthesis, ⇢ indicates indirect synthesis.

**Figure 7 ijms-25-01663-f007:**
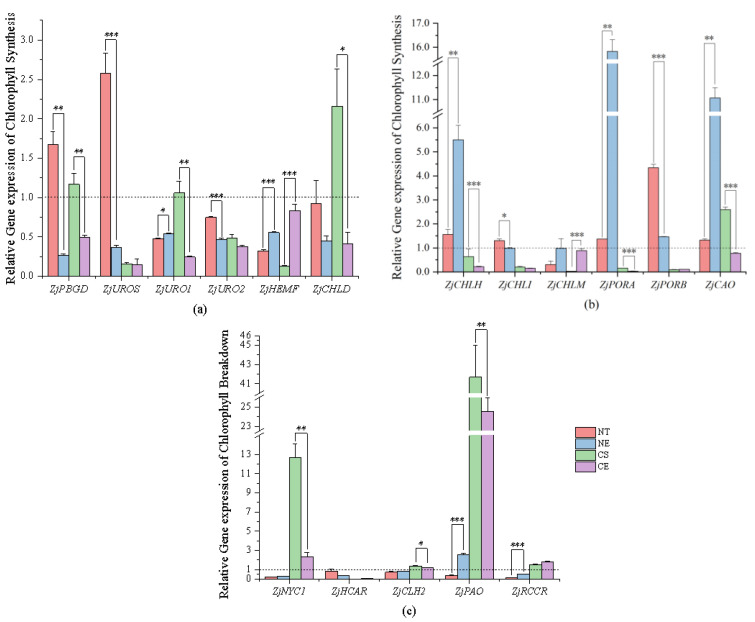
Relative gene expression in chlorophyll metabolism; (**a**) relative gene expression (*ZjPBGD*, *ZjUROS*, *ZjURO1*, *ZjURO2*, and *ZjHemF*) in the process of chlorophyll synthesis in different *Z. japonica* plants on 52 d compared to 0 d; (**b**) relative gene expression (*ZjCHLD*, *ZjCHLH*, *ZjCHLI*, *ZjChlM*, *ZjPORA*, and *ZjPORB*) in the process of chlorophyll synthesis in different *Z. japonica* plants on 52 d compared to 0 d; and (**c**) relative gene expression (*ZjCAO*, *ZjNYC1*, *ZjHCAR*, *ZjCLH2*, *ZjPAO*, and *ZjRCCR*) in the process of chlorophyll breakdown in different *Z. japonica* plants on 52 d to 0 d. ‘*’, ‘**’, and ‘***’ indicate significant differences at *p* < 0.05; *p* < 0.01; and *p* < 0.001, respectively. The dashed line refers to the relative gene expression is 1.

**Figure 8 ijms-25-01663-f008:**
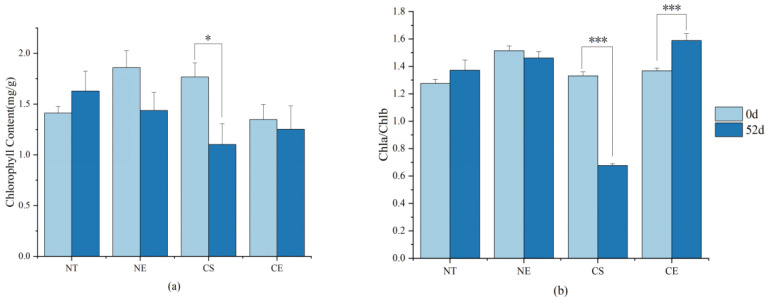
Chlorophyll content in different *Z. japonica* plants on 0 d and 52 d; (**a**) chlorophyll content in different *Z. japonica* plants on 0 d and 52 d; and (**b**) Chl *a*/*b* in different *Z. japonica* plants on 0 d and 52 d. ‘*’, and ‘***’ indicate significant differences at *p* < 0.05 and *p* < 0.001, respectively. NT (control) refers to normal day/night temperatures (32/28 °C) with water pretreatment as a control; NE refers to normal day/night temperatures (32/28 °C) with ethephon pretreatment (150 mg/L); CS refers to day/night cold stress (6/4 °C) with water pretreatment; and CE refers to day/night cold stress (6/4 °C) with ethephon pretreatment (150 mg/L).

**Figure 9 ijms-25-01663-f009:**
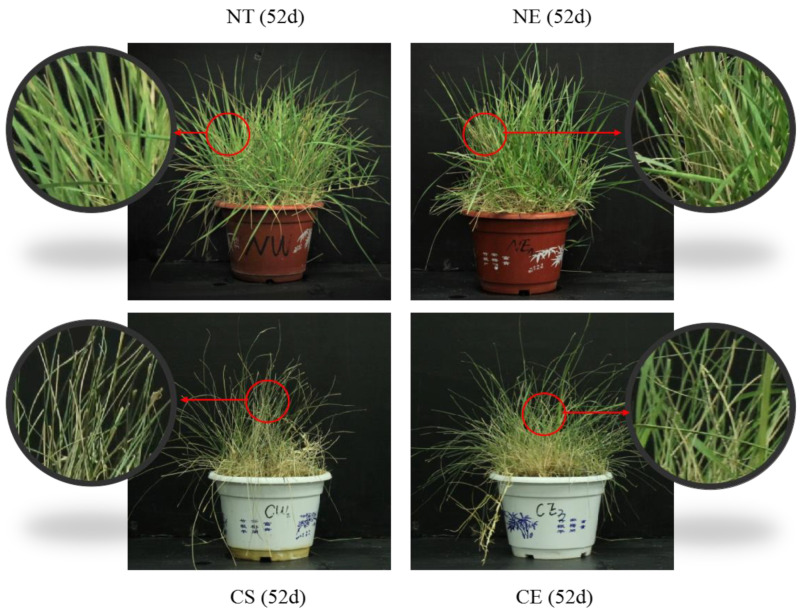
Effects of ethephon pretreatment under non-stressed conditions and under cold stress in *Z. japonica*. NT (control) refers to normal day/night temperatures (32/28 °C) with water pretreatment as a control; NE refers to normal day/night temperatures (32/28 °C) with ethephon pretreatment (150 mg/L); CS refers to day/night cold stress (6/4 °C) with water pretreatment; and CE refers to day/night cold stress (6/4 °C) with ethephon pretreatment (150 mg/L).

**Figure 10 ijms-25-01663-f010:**
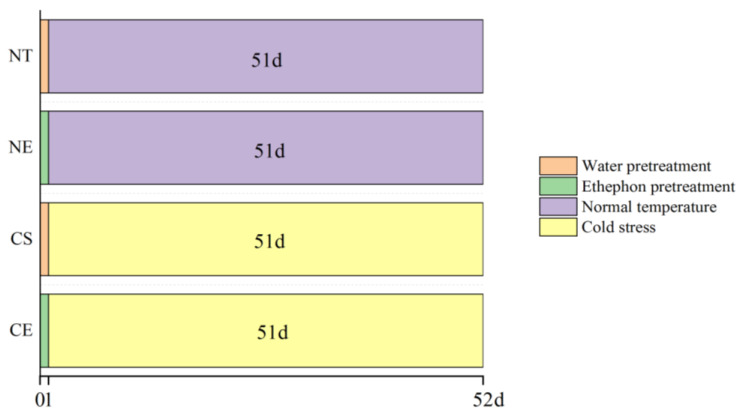
Treatment schedule for *Z. japonica* plants at normal temperature and under cold stress. NT (control), normal day/night temperatures (32/28 °C) with water pretreatment as a control; NE, normal day/night temperatures (32/28 °C) with ethephon pretreatment (150 mg/L); CS, day/night cold stress (6/4 °C) with water pretreatment; and CE, day/night cold stress (6/4 °C) with ethephon pretreatment (150 mg/L).

## Data Availability

Data are contained within the article or Appendix A.

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
