# Peer review of "The Effect of Ethephon on Ethylene and Chlorophyll in Zoysia japonica Leaves"

_ijms, 2024, doi:10.3390/ijms25031663_

Round 1
Reviewer 1 Report
Comments and Suggestions for Authors
Though the article has been written accordingly with the interest to the readers there are many typo errors in the manuscript.
Give the references in materials and methods
Re write 4.3, 4.7 in past tense
Format the references according to the journal
Keep the botanical name of the plant in italics eg: line no 130, etc.
Write a paragraph in Materials and methods for statistical analysis of the data
Give the abbreviation in figure 4. Also write the figure 4 in non-italics
Complete the bracket in line no: 494
Briefly elaborate on the importance and future scope of the study
Comments on the Quality of English LanguageMinor English language editing is required
Reviewer 2 Report
Comments and Suggestions for Authors
Dear Authors,
The manuscript presents orginal results, however it should be revised for the quality with some recommendations.
Sincerely yours,

The english language should be controlled.
Reviewer 3 Report
Comments and Suggestions for Authors
Title: Ethephon plays a positive role in releasing ethylene and maintaining chlorophyll content in Zoysia japonica leaves
Authors studied in this work the effect of ethephon application on the warm-season turfgrass (Zoysia japonica) under cold stress. Their results show that ethephon improves plant growth and tolerance under cold stress. Also, gene expression results, determined the upregulation of enzyme activities and genes implicated in ethylene biosynthesis as well their induction rates, and chlorophyll metabolism (A/B). The general results introduce ethephon as a biostimulator product for promoting plant growth and tolerance to cold stress in Z. japonica.
The paper is very interesting and very well written and discussed. I recommend it for publication at the presence state with a recommendation to modify the title to “Ethephon plays a positive role in releasing ethylene and maintaining chlorophyll content in Zoysia japonica leaves, promoting plant growth and cold tolerance”
Round 2
Reviewer 2 Report
Comments and Suggestions for Authors
Dear Authors,
Thank you very much for your revised manuscript.
I think that it is so good now.